# Comprehensive Study of the IBMP ELISA IgA/IgM/IgG COVID-19 Kit for SARS-CoV-2 Antibody Detection

**DOI:** 10.3390/diagnostics14141514

**Published:** 2024-07-13

**Authors:** Sibelle Botogosque Mattar, Paola Alejandra Fiorani Celedon, Leonardo Maia Leony, Larissa de Carvalho Medrado Vasconcelos, Daniel Dias Sampaio, Fabricio Klerynton Marchini, Luis Gustavo Morello, Vanessa Hoysan Lin, Sandra Crestani, Aquiles Assunção Camelier, André Costa Meireles, André Luiz Freitas de Oliveira Junior, Antônio Carlos Bandeira, Yasmin Santos Freitas Macedo, Alan Oliveira Duarte, Tycha Bianca Sabaini Pavan, Isadora Cristina de Siqueira, Fred Luciano Neves Santos

**Affiliations:** 1Molecular Biology Institute of Paraná (IBMP), Curitiba 81350-010, PR, Brazil; sibelle.mattar@ibmp.org.br (S.B.M.); paola.fiorani@fiocruz.br (P.A.F.C.); fabricio.marchini@fiocruz.br (F.K.M.); luis.morello@fiocruz.br (L.G.M.); vanessa.lin@ibmp.org.br (V.H.L.); sandra.crestani@ibmp.org.br (S.C.); 2Interdisciplinary Research Group in Biotechnology and Epidemiology of Infectious Diseases (GRUPIBE), Gonçalo Moniz Institute, Oswaldo Cruz Foundation-Bahia (FIOCRUZ-BA), Salvador 402596-710, BA, Brazil; leonardo.leony@fiocruz.br (L.M.L.); larissa.vasconcelos@fiocruz.br (L.d.C.M.V.); diassampaio@gmail.com (D.D.S.); tycha.pavan@fiocruz.br (T.B.S.P.); isadora.siqueira@fiocruz.br (I.C.d.S.); 3Advanced Public Health Laboratory, Gonçalo Moniz Institute, Oswaldo Cruz Foundation-Bahia (FIOCRUZ-BA), Salvador 402596-710, BA, Brazil; 4Laboratory for Applied Science and Technology in Health, Carlos Chagas Institute, Oswaldo Cruz Foundation-Paraná (FIOCRUZ-PR), Curitiba 81350-010, PR, Brazil; 5Aliança D’Or Hospital, Salvador 41920-180, BA, Brazil; aquilescamelier@yahoo.com.br (A.A.C.); andremeireles001@gmail.com (A.C.M.); andreluizfojr@gmail.com (A.L.F.d.O.J.); 6Aeroporto Hospital, Lauro de Freitas 42700-000, BA, Brazil; antoniobandeira@gmail.com; 7Laboratory of Investigation in Global Health and Neglected Diseases, Gonçalo Moniz Institute, Oswaldo Cruz Foundation-Bahia (FIOCRUZ-BA), Salvador 402596-710, BA, Brazil; yasmin.macedo@fiocruz.br (Y.S.F.M.); alanduarte.ufba@gmail.com (A.O.D.); 8Integrated Translational Program in Chagas Disease from FIOCRUZ (Fio-Chagas), Oswaldo Cruz Foundation-Rio de Janeiro (FIOCRUZ-RJ), Rio de Janeiro 21040-360, RJ, Brazil

**Keywords:** COVID-19, immunodiagnosis, SARS-CoV-2, sensitivity, specificity

## Abstract

COVID-19 laboratory diagnosis primarily relies on molecular tests, highly sensitive during early infection stages with high viral loads. As the disease progresses, sensitivity decreases, requiring antibody detection. Since the beginning of the pandemic, serological tests have been developed and made available in Brazil, but their diagnostic performance varies. This study evaluated the IBMP ELISA IgA/IgM/IgG COVID-19 kit performance in detecting SARS-CoV-2 antibodies. A total of 90 samples, including 64 from COVID-19 patients and 26 pre-pandemic donors, were assessed based on time post symptom onset (0–7, 8–14, and 15–21 days). The kit showed 61% sensitivity, 100% specificity, and 72% accuracy overall. Sensitivity varied with time, being 25%, 57%, and 96% for 0–7, 8–14, and 15–21 days, respectively. Similar variations were noted in other commercial tests. The Gold ELISA COVID-19 (IgG/IgM) had sensitivities of 31%, 71%, and 100%, while the Anti-SARS-CoV-2 NCP ELISA (IgG) and Anti-SARS-CoV-2 NCP ELISA (IgM) showed varying sensitivities. The IBMP ELISA kit displayed high diagnostic capability, especially as the disease progressed, complementing COVID-19 diagnosis. Reproducibility assessment revealed minimal systematic and analytical errors. In conclusion, the IBMP ELISA IgA/IgM/IgG COVID-19 kit is a robust tool for detecting anti-SARS-CoV-2 antibodies, increasing in efficacy over the disease course, and minimizing false negatives in RT-PCR COVID-19 diagnosis.

## 1. Introduction

SARS-CoV-2, a single-stranded positive RNA virus, belongs to the coronavirus family, which also includes the viruses responsible for severe acute respiratory syndrome (SARS) and Middle East respiratory syndrome (MERS). It is believed to have originated in bats [1] and may have transmitted to humans through an intermediate host, such as a pangolin or civet cat [2,3,4]. The virus is highly contagious, primarily spreading through respiratory droplets when an infected person talks, coughs, or sneezes. Transmission can also occur through close contact, such as touching surfaces contaminated with the virus and then touching the mouth, nose, or eyes. COVID-19, caused by this virus, manifests with a range of symptoms, from mild to severe, including fever, cough, fatigue, body aches, and breathing difficulties. While many recover without complications, severe illness or death can occur, particularly among the elderly and those with underlying health conditions [5,6,7]. The disease first emerged in Wuhan, China, in December 2019 [8] and was declared a pandemic by the World Health Organization (WHO) on 11 March 2020, due to the rapid increase in cases and wide geographic spread [9]. As of 5 March 2023, WHO no longer considers COVID-19 a public health emergency of international concern [10].

Diagnosing COVID-19 relies on a combination of clinical symptoms, epidemiological characteristics, and laboratory tests [11]. Reverse transcriptase-polymerase chain reaction (RT-PCR) is the most used test due to its high sensitivity and specificity in detecting the virus at the disease’s onset [12]. Specimen collection, usually through nasal or throat swabs or saliva, is straightforward but prone to errors and contamination. Additionally, RT-PCR is labor-intensive and costly, and requires specialized laboratory facilities and personnel, making it challenging for point-of-care use. An alternative is antigen testing, which detects specific SARS-CoV-2 proteins. While less sensitive than molecular tests, antigen tests provide rapid results in 15–30 min and are suitable for self-testing. They meet WHO’s COVID-19 diagnosis criteria (sensitivity ≥80% and specificity ≥97%) and can replace laboratory-based RT-PCR in urgent patient care situations or when timely RT-PCR is unavailable [13]. However, both molecular and antigen tests are effective in the early infection stage when viral loads are high, but their sensitivity decreases as the disease progresses [14,15] or in the presence of SARS-CoV-2 variants [16,17], prompting the need for indirect immunoassays to detect virus-specific antibodies [18].

Antibody tests readily detect anti-SARS-CoV-2 antibodies in serum a few days to weeks after infection or vaccination but do not diagnose active COVID-19 cases [19,20]. They are valuable for estimating virus prevalence and identifying potential plasma donors [21,22]. Antibody tests are relatively simple to perform and do not require extensive training, making them suitable for various settings, including clinics, hospitals, and home use [23]. Since the pandemic’s onset, several commercial antibody tests have become available in Brazil. One of these is the IBMP ELISA IgA/IgM/IgG COVID-19 kit, designed to detect antibodies against the virus’s spike antigen in serum or plasma samples. Despite its availability in the Institute of Molecular Biology of Paraná’s (IBMP) portfolio, there has been no systematic evaluation of its performance. In light of this scenario, our study aims to critically assess its diagnostic performance in detecting anti-spike antibodies of SARS-CoV-2 and compare its performance with kits commonly used in Brazil, such as Euroimmun IgG, Euroimmun IgM, and Gold ELISA COVID-19 IgG + IgM.

## 2. Materials and Methods

### 2.1. Study Design

We determined the required sample size for assessing performance using the open-source software OpenEpi (The OpenEpi Project, Atlanta, GA, USA) [24]. Assuming an infinite population, a 95% confidence interval, a 2.5% absolute error, and an expected sensitivity of 99%, we estimated that a minimum of 61 sera from individuals infected with SARS-CoV-2 would be needed. In total, we obtained 90 sera samples, including 26 from SARS-CoV-2-negative individuals and 64 from SARS-CoV-2-positive individuals. All positive samples were from unvaccinated patients who had tested positive for SARS-CoV-2 RT-PCR and exhibited clinical symptoms consistent with COVID-19. These samples were collected from patients treated at hospitals in the city of Salvador, Bahia, and surrounding areas, namely, Aliança D’Or and Aeroporto Hospitals (Figure 1), between March and October 2020. We categorized the SARS-CoV-2 patient samples based on the time of symptom onset into the following groups: 0–7 days post symptom onset (week 1), 8–14 days post symptom onset (week 2), and 15–21 days post symptom onset (week 3) [14]. Negative samples were sourced from the Hematology and Hemotherapy Foundation of Bahia (HEMOBA) and were obtained from healthy individuals in the pre-pandemic period who had tested negative for Chagas disease, HBV, HCV, HIV-1/2, HTLV-1/2, and syphilis (Figure 1). To assess imprecision, interferents, and cross-reactivity, we utilized a convenience sample size. Additionally, we employed a commercial panel (Access Biologicals, Vista, CA, USA; catalogue number 16080548) to evaluate cross-reactivity, comprising 21 samples from patients with unrelated diseases, as defined by their serological or parasitological diagnoses. These unrelated diseases encompassed both infectious diseases such as adenovirus, cytomegalovirus, dengue, Epstein–Barr virus, hepatitis B virus, hepatitis C virus, herpes simplex, HIV-1, HIV-2, influenza, Lyme disease, parainfluenza 1, 2, and 3, parvo-B19, respiratory syncytial virus, rubella, syphilis, toxoplasmosis, varicella-zoster, and Zika virus, and autoimmune diseases (anti-double-stranded DNA autoantibodies, antinuclear antibodies, myeloma, and rheumatoid factor). It is important to note that only one sample was available for each unrelated disease within the commercial panel.

### 2.2. IBMP ELISA IgA/IgM/IgG COVID-19

All serum samples were tested for the presence or absence of anti-spike IgA/IgM/IgG antibodies using the IBMP Spike IgA/IgM/IgG ELISA (IBMP, Curitiba, PR, Brazil), following the manufacturer’s instructions. The samples were diluted 1:120 in sample buffer, distributed along with the calibrator and controls (100 µL each) onto a microplate assay, and incubated at 37 °C for 30 min. After washing five times with a 1:10 diluted wash buffer, 100 µL of enzyme conjugate was added, followed by another incubation under the same conditions. The liquid was then removed, and 80 µL of chromogenic substrate was introduced. The reactions were halted after a 10 min incubation at room temperature in the dark, and the results were obtained using a microplate spectrophotometer (SPECTRAmax 340PC^®^, Molecular Devices Corporation, San José, CA, USA or BioTek SynergyH1, BioTek Instruments Inc., Winoosk, VT, USA) at 450 nm.

### 2.3. Imprecision Assessment

We assessed imprecision through two types of testing: within-run (repeatability) and intra-lab testing. Within-run testing encompassed 64 SARS-CoV-2 positive and 26 negative samples. Each sample underwent two sequential assessments within the same run, labeled as Replicate 1 and Replicate 2. Intra-lab testing, spanning three different batches, involved six samples: two negatives for SARS-CoV-2 and four positives for SARS-CoV-2. The intra-lab testing considered repeatability, between runs, within days, and within lab evaluations conducted on consecutive days. All these analyses adhered to the guidelines outlined in CLSI guideline EPS5-A2 [25].

### 2.4. Interferants Analysis

We investigated interference from bilirubin, triglycerides, and hemoglobin in the IBMP Spike IgA/IgM/IgG ELISA for anti-SARS-CoV-2 antibody detection. Following the manufacturer’s instructions, we diluted the serum reagent and prepared interferent solutions. Hemoglobin (Merck, catalogue number H7379) was reconstituted to 20 mg/mL in ultrapure water, bilirubin (Merck, catalogue number 14370) in a 0.1 M NaOH solution (pH 9.0), and triglycerides (Lee Biosolutions, catalogue number 361.56-0.01) to 120 mg/mL using 0.15 M NaCl. The serum reagent included a SARS-CoV-2-positive sample (RI > 2.0) and a negative sample (RI < 0.06), both diluted with interferent solutions at a 1:1 ratio for the desired concentrations. Precision was measured as the coefficient of variation (CV%).

### 2.5. Commercial ELISA Test Comparison

We selected all SARS-CoV-2-positive samples (*n* = 64) to compare the performance and agreement between the IBMP Spike IgA/IgM/IgG ELISA and commercial SARS-CoV-2 ELISA tests. The commercial tests were selected based on their commercial availability and approval for use in Brazil. Specifically, we included the following three commercial COVID-19-specific enzyme immunoassays: (1) Anti-SARS-CoV-2 NCP IgM ELISA (Euroimmun Medizinische Labordiagnostika AG, Lübeck, Germany), which detects IgM antibodies against SARS-CoV-2 using the nucleocapsid (N) of the virus; (2) Anti-SARS-CoV-2 IgG ELISA (Euroimmun Medizinische Labordiagnostika AG, Lübeck, Germany), which detects IgG antibodies to SARS-CoV-2 using the S1 domain of the spike protein, including the immunologically relevant receptor binding domain (RBD); and (3) GOLD ELISA COVID-19 IgG + IgM (REM Diagnóstica, São Paulo, SP, Brazil), which detects both IgG and IgM using the S1 and S2 domains of the spike protein and the N protein. All tests were conducted following the manufacturer’s instructions.

### 2.6. Statistical Analysis

Data were encoded, analyzed, and presented as scatter plots using GraphPad Prism software (GraphPad Prism version 10.0.0, San Diego, CA, USA). Descriptive statistics are expressed as geometric mean ± standard deviation (SD). The normality of the dataset was tested with the Shapiro–Wilk test, followed by Student’s *t*-test, and when the homogeneity assumption could not be confirmed, the Wilcoxon signed-ranks test was applied. All analyzes were two-sided tests, and a *p*-value of less than 5% was considered significant (*p*-value < 0.05). We used cut-off point analysis to determine the optimal optical density value (OD) for distinguishing between negative and positive samples, following the manufacturer’s instructions. Results were presented as an index format, representing the ratio between OD of a given sample and the cut-off OD for each microplate, referred to as the reactivity index (RI). Samples with an RI < 1.00 were considered negative. To assess the overall accuracy of the IBMP Spike IgA/IgM/IgG ELISA kit, we calculated the area under the receiver operating characteristic curve (AUC), with classification as outstanding (1.0), elevated (0.82–0.99), moderate (0.62–0.81), or low (0.51–0.61) [26]. Furthermore, we evaluated ELISA performance using a dichotomous approach and compared it in terms of sensitivity (SEN), specificity (SPE), accuracy (ACC), likelihood ratio (LR), and diagnostic odds ratio (DOR) [27]. Accuracy reflects the test’s ability to produce correct results, while LR, whether negative or positive, indicates the likelihood of tested individuals having a positive or negative result compared to untested individuals. Positive LRs above 10 and negative LRs below 0.1 are generally considered significant for diagnosis [28]. For a test with 100% specificity, determining positive LR becomes mathematically impossible because the formula uses the “1-specificity” equation as the denominator. A strategy to overcome this problem is to subtract a value from the sensitivity and specificity values to obtain an estimated result for the positive LR [29,30]. DOR was calculated as the ratio between positive and negative LR values and serves as a comprehensive parameter summarizing the diagnostic accuracy of the test. It represents the probability of obtaining a positive result in a person with the disease compared to a person without the disease [31]. We used confidence intervals (CI) with a confidence level of 95% (95%CI), and non-overlapping 95% CI bars indicated statistical significance [32]. The strength of agreement between the results of the IBMP Spike IgA/IgM/IgG ELISA and the reference test used in the study (clinical presentation consistent with COVID-19 + positive RT-PCR) was assessed using Cohen’s kappa coefficient (k) [33] interpreted as follows: almost perfect (0.81 < k ≤ 1.0), substantial (0.61 < k ≤ 0.80), moderate (0.41 < k ≤ 0.60), fair (0.21 < k ≤ 0.40), slight (0 < k ≤ 0.20), and poor (k = 0) agreement. The within-run reproducibility of immunoassays was assessed using the Bland–Altman method and Deming regression, performed with MedCalc^®^ Statistical Software version 20.218 (MedCalc Software, Ltd., Ostend, Belgium). The Bland–Altman plot with limits of agreement (LoAs) evaluated the variability and magnitude between Replicate 1 and Replicate 2 [34]. Deming regression was used to mathematically determine the agreement between replicates of the IBMP Spike IgA/IgM/IgG ELISA as well as proportional bias (slope, 95% CI) and systematic bias (intercept, 95% CI). The Deming regression analysis yielded a null hypothesis when the intercept and slope were 0 and 1, respectively. Intra-lab testing was assessed using standard deviation (SD) and the coefficient of variation (CV). We prepared a flowchart (Figure 1) in accordance with the STARD guidelines (Standards for the Reporting of Diagnostic Accuracy Studies) [35].

## 3. Results

### 3.1. Characteristics of Participants

In this study, we included 90 anonymized unvaccinated human serum samples that were collected previously. Among the 64 individuals who tested positive for SARS-CoV-2, the average age was 52 years, with an interquartile range (IQR) of 40.0 to 66.0 years, and the female-to-male ratio was 1/2.4. On admission to the hospital, the most common signs and symptoms were fever (45/64; 70.3%), a nonproductive cough (51/64; 79.7%), and dyspnea (48/64; 75%). Upon evaluation by the medical team, 68.8% (44/64) of patients were admitted to an intensive care unit (ICU), while 31.2% (20/64) were treated as outpatients and subsequently discharged. Among those admitted to the ICU, 84.1% (37/44) recovered and were discharged, while 15.9% (7/44) did not survive. For individuals who tested negative for SARS-CoV-2 (*n* = 26), the female-to-male ratio was approximately 1 to 1.2. All blood donors came from the state of Bahia, with distribution as follows: Salvador (15.4%), Alagoinhas (30.8%), Aporá (7.7%), Catu (19.2%), and Lauro de Freitas (26.9%). No age information for this group is available.

### 3.2. Diagnostic Performance of IBMP Spike IgA/IgM/IgG ELISA

The diagnostic performance of the IBMP Spike IgA/IgM/IgG ELISA is illustrated in Figure 2. For this analysis, SARS-CoV-2-positive samples were analyzed without stratification by symptom onset, providing a global assessment of assay performance. The assay generated a reactivity index (RI) value of 1.50 for positive samples, significantly higher than the RI of 0.24 for negative samples (Figure 2A). Among the positive samples, 22 were classified as negative (IR < 0.80), 4 as indeterminate (IR ≥ 0.80 and <1.10), and 38 as positive (IR ≥ 1.10), resulting in a sensitivity of 60.9%. Negative samples showed reactivity indices below 0.80, indicating a specificity of 100%. The area under the curve (AUC) analysis yielded a value of approximately 0.83 (*p* < 0.0001), indicating elevated overall ability of the assay to correctly distinguish between positive and negative serum samples (Figure 2B). As shown in Figure 2C, the accuracy of the IBMP Spike IgA/IgM/IgG ELISA was estimated to be 72.2%, and the diagnostic odds ratio (DOR) value, based on likelihood ratios, was 407.2. The qualitative evaluation of the results using Cohen’s kappa method showed substantial agreement (k = 0.72) between the reference test (clinical presentation + RT-PCR) and the IBMP Spike IgA/IgM/IgG ELISA.

We also assessed the performance of the IBMP Spike IgA/IgM/IgG ELISA in different infection periods, classifying samples based on symptom onset as follows: 0–7 days post symptom onset (*n* = 20; Median: 4 days; IQR: 2.25–5 days), 8–14 days post symptom onset (*n* = 21; Median: 10 days; IQR: 9–13 days), and 15–21 days post symptom onset (*n* = 23; Median: 18 days; IQR: 16–21 days). The same set of negative SARS-CoV-2 samples was used to calculate performance parameters requiring negative samples, such as specificity, accuracy, likelihood ratio, and DOR.

As depicted in Figure 3 (Appendix A), levels of anti-spike antibody in SARS-CoV-2 positive samples exhibited a significant increase between days 0–7 (RI = 0.41), 8–14 (RI = 1.47; *p* = 0.0004), and 15–21 (RI = 4.31; *p* < 0.0001), as well as between days 8–14 and 15–21 (*p* = 0.0002). This increase in antibody levels led to higher sensitivity values and other diagnostic parameters, including accuracy, DOR, and Cohen’s kappa method. Sensitivity values progressed from 25% in the initial days of infection to 57.1% during days 8–14 and ultimately reached 95.7% in the third week of infection. In comparison to the first 0–7 days, the DOR values increased 4-fold in the second period and 66-fold in the last period. Between days 8–14 and 15–21, DOR increased 16.5-fold. Notably, a low AUC value was obtained during days 0–7 (AUC = 51.9%), whereas elevated values were observed on days 8–14 (AUC = 97%) and 15–21 (AUC = 99.8%).

Figure 4 provides a visual representation of the repeatability evaluation of IBMP Spike IgA/IgM/IgG ELISA, utilizing Deming regression analysis (left panel) and the Bland–Altman plot (right panel). The results reveal a strong agreement between Replicate 1 and Replicate 2, as indicated by an R-squared value of 0.97, an intercept of 0.02762 (95%CI, −0.02060 to 0.07584), and a slope of 1.0413 (95%CI, 0.9733 to 1.1093), demonstrating minimal systematic and random errors. In the Bland–Altman analysis, the mean bias was 5.1% (95%CI, 1.4 to 8.8%), and LoA values ranged from −29.7% to 39.8%. Although the line of equality did not fall within the 95% confidence interval of bias, the discrepancy between replicates was deemed nonsignificant, supported by a *p*-value for the null hypothesis of less than 0.05. Only two points (2.2%) deviated outside the LoAs, aligning with the expected 5% tolerance. Furthermore, the agreement in within-run imprecision was perfect, signified by a strength of agreement of 1.00. The coefficient of variation (CV) for SARS-CoV-2-positive and -negative samples was 8.7% and 10.9%, respectively.

The reliability of diagnostic assays relies on the consistency of their components. In this context, we assessed assay imprecision through comparative analyses of experiments conducted within a single run, between runs, within days, and within laboratory, utilizing a serum panel containing well-defined samples from both SARS-CoV-2-negative (*n* = 2; Table 1: samples #1 and #2) and -positive (*n* = 4; Table 1: samples #3 to #6) cases. To measure imprecision, we employed the standard deviation (SD) and coefficient of variation (CV) as statistical parameters. Table 1 provides a summary of SD and CV analysis for each dataset. Across all analyses, the CV for each batch of positive samples fell within a 20% range, indicating acceptable reproducibility. There were two exceptions in the between-runs analysis, with sample #6 from batch two registering 22.44% and sample #4 from batch three recording 23.93%. Conversely, SARS-CoV-2-negative samples, except for repeatability, exhibited CV values exceeding 20%, ranging from 25.62% to 47.11%. Despite this variability, it is important to note that none of the samples produced a false positive result during the second run.

The potential cross-reactivity (RI ≥ 1.0) of the IBMP Spike IgA/IgM/IgG ELISA was assessed using serum samples from 21 individuals with unrelated diseases. The average RI consistently remained low for all diseases, with an average of 0.19 and a range from 0.07 to 1.04. Notably, it is worth highlighting that, apart from a sample testing positive for syncytial respiratory virus, which displayed a high RI intensity and resulted in an inconclusive result (RI = 1.04), all other cross-reacting samples exhibited relatively low RI intensities.

Substance interference had no significant impact on the signal of the IBMP Spike IgA/IgM/IgG ELISA kit, as illustrated in Figure 5. The coefficient of variation (CV%) for the SARS-CoV-2-positive samples remained below 10% across varying concentrations of bilirubin (0.117–0.7 mg/mL) and hemoglobin (1.0–7.5 mg/mL). Triglycerides exhibited similar trends, with slightly higher CV observed at 20 mg/mL (still below 20%, as anticipated for immunoassays). Negative results showed no variation.

We conducted a comparison of the results obtained from SARS-CoV-2 positive samples using the IBMP Spike IgA/IgM/IgG ELISA with three other commercially available kits in Brazil. In the overall performance analysis (Figure 6), sensitivity values varied, with the Anti-SARS-CoV-2 IgG ELISA at 51.6%, the IBMP Spike IgA/IgM/IgG ELISA and the Anti-SARS-CoV-2 NCP IgM ELISA at 60.9%, and the GOLD ELISA COVID-19 IgG/IgM at 67.2%. The Anti-SARS-CoV-2 IgG ELISA had the lowest RI value (RI = 0.85), which fell below the cut-off (RI = 1.00). However, due to the overlap in 95%CI, no significant differences were observed in sensitivity and RI among the kits. Similar results were found when samples were categorized by days post symptom onset. In all four tests, diagnostic performance improved with the time of infection, and sensitivity increased over time, reaching 87% for the Anti-SARS-CoV-2 NCP IgM ELISA and the Anti-SARS-CoV-2 IgG ELISA, 95.7% for the IBMP Spike IgA/IgM/IgG ELISA, and 100% for the GOLD ELISA COVID-19 IgG/IgM. Once again, no significant differences were observed in sensitivity and RI among them (Figure 6).

In order to understand the behavior of the IBMP Spike IgA/IgM/IgG ELISA and other commercial tests during COVID-19 seroconversion, we analyzed the arithmetic mean of the reactivity index in relation to days post symptom onset (Figure 7). The GOLD ELISA COVID-19 IgG/IgM was the first test to detect seroconversion at 4–5 days post symptom onset, followed by the IBMP Spike IgA/IgM/IgG ELISA on day 6, the Anti-SARS-CoV-2 IgG ELISA on day 7, and the Anti-SARS-CoV-2 IgM ELISA on day 9 post symptom onset.

## 4. Discussion

The rapid development and implementation of diagnostic tests for emerging diseases pose significant challenges due to a lack of adequate diagnostic evaluation, resources, and logistical constraints associated with an outbreak. Given the public health urgency of the COVID-19 pandemic, many diagnostic tests were rapidly produced and deployed without an adequate analysis of their diagnostic value [36]. With this in mind, we conducted a careful evaluation of the performance of the new IBMP ELISA IgA/IgM/IgG COVID-19 kit for the detection of anti-SARS-CoV-2 antibodies in sera from individuals with COVID-19. The results described here indicate a significant ability of the test to discriminate between SARS-CoV-2-positive and -negative samples from the second week post symptom onset, with high sensitivity values and similar performance to other tests commercially available in Brazil.

Global performance evaluation was performed using the IBMP ELISA IgA/IgM/IgG COVID-19 kit to determine diagnostic sensitivity, specificity, and accuracy for COVID-19. Despite the fact that the sensitivity value was less than 70%, no difference was observed compared to other commercial kits. Although not significant, a higher sensitivity value was obtained with the GOLD ELISA COVID-19 IgG/IgM. In this assay, a mixture of three recombinant proteins of SARS-CoV-2 (S1 + S2 + N) is used to detect IgM and IgG against the virus. In contrast, the IBMP ELISA IgA/IgM/IgG COVID-19 kit uses only one recombinant protein (spike, whole molecule) to detect IgG and IgM antibodies in addition to IgA (not used in the other kits evaluated here). The presence of IgA in human milk suggests that this biological material could serve as an alternative for COVID-19 diagnosis [37,38]. The Anti-SARS-CoV-2 NCP IgM ELISA provided the same sensitivity value as the IBMP ELISA IgA/IgM/IgG COVID-19 kit. This kit uses the nucleocapsid (N) of the virus to detect anti-SARS-CoV-2 IgM. The Anti-SARS-CoV-2 IgG ELISA provided the lowest sensitivity value. This kit uses the S1 domain of the spike protein to detect IgG against SARS-CoV-2. In the global performance analysis, the use of samples at the onset of infection was responsible for a high proportion of SARS-CoV-2-positive samples misclassified as negative (39.1%), resulting in an accuracy value of 72.2%. However, accuracy reached 80.9% and 97.8% for samples from individuals tested 8–14 and 15–21 days post symptom onset, respectively.

In this study, we observed an increase in reactivity index and sensitivity values for the IBMP Spike IgA/IgM/IgG ELISA with time post infection. Indeed, 12 SARS-CoV-2-positive samples were falsely classified as negative, and sensitivity reached a value of 25% at week 1 post infection, followed by 6 samples at week 2 and a sensitivity of 57.1% and 1 sample at week 3 and a sensitivity of 95.7%. These values were similar to the GOLD ELISA COVID-19 IgG/IgM, where sensitivity ranged from 30% to 66.7% in the first two weeks and 100% in the third infection period. On the other hand, the Anti-SARS-CoV-2 IgG ELISA and SARS-CoV-2 NCP IgM ELISA kits provided lower sensitivity values compared to the IBMP ELISA IgA/IgM/IgG COVID-19 kit and the GOLD ELISA COVID-19 IgG/IgM. In the literature, sensitivity values for the Anti-SARS-CoV-2 IgG ELISA vary from 37.8% to 93.3% [39,40,41,42], with a mean sensitivity of 74.4% (CI95% 66.9–82%) [43], a higher sensitivity value compared to our results. Several authors have also reported an increase in the signal of indirect immunoassays and sensitivity of kits using sera from patients at different time points during COVID-19. These authors noted a significant increase in the diagnostic performance of the assays when samples from patients with more than 10 days of symptomatology were used in the evaluations [14,40,41]. Surprisingly, the SARS-CoV-2 NCP IgM ELISA kit for the detection of IgM antibodies at the onset of infection had the same sensitivity as the Anti-SARS-CoV-2 IgG ELISA, a specific kit for the detection of IgG.

Evaluating diagnostic tests based on sensitivity, specificity, and accuracy is insufficient to measure their impact on clinical decisions. A diagnostic test is useful only if its results alter the probability of disease occurrence. The determination of the likelihood ratio (LR) is useful to describe the discriminatory power of a test and defines the probability of a given result in infected individuals versus the probability of the same result in healthy individuals [28]. In this study, the IBMP ELISA IgA/IgM/IgG COVID-19 kit showed a specificity of 100%. Therefore, we subtracted 0.05 from the sensitivity and specificity values to obtain an estimated positive LR of 159.7, which means that a SARS-CoV-2-positive individual has approximately 160-fold higher probability of having COVID-19 when tested with the IBMP ELISA IgA/IgM/IgG COVID-19 kit. A higher positive likelihood ratio was observed in samples from individuals 15–21 days post symptom onset. SARS-CoV-2-negative samples yielded LR values of 0.39, 0.75, and 0.43 in global analysis and 0–7 and 8–14 days post symptom onset, respectively. However, a value of 0.04 resulted 15–21 days post symptom onset. It is agreed that negative LRs below 0.1 and positive LRs above 10 contribute significantly to the diagnosis [44]. DOR describes the probability of obtaining a positive result for a person with an infection, as opposed to someone who is not infected [31]. It is a universal performance parameter that summarizes diagnostic test accuracy. Here, we observed values above 407, which means that a person tested with the IBMP ELISA IgA/IgM/IgG COVID-19 kit is ~407 times more likely to receive a correct diagnosis than a person who is not tested. At week 3 (15–21 days post-symptom onset), DOR reached a value of 5742, indicating high applicability in disease progression.

Part of the process of validating a method refers to the assessment of precision, which can be defined as the degree of agreement between independent measurements performed under identical conditions. While precision refers to the concept of variation around a central value, imprecision is what is actually measured. Here, two statistical approaches were used to examine within-run imprecision using SARS-CoV-2-positive and -negative samples tested in replicate. Deming regression analysis showed that the IBMP ELISA IgA/IgM/IgG COVID-19 kit has no proportional bias, indicating that this assay is in complete agreement for SARS-CoV-2-positive and -negative samples, as shown by the Bland–Altman plot. This result indicates the high linearity of Replicate 2 compared with Replicate 1. The regression analysis showed no systematic negative bias, indicating that the results obtained with the replicates are similar. No difference was observed between the replicates in the diagnostic performance parameters in terms of sensitivity, specificity, and accuracy either. The qualitative evaluation of the results showed perfect agreement using Cohen’s kappa method. For both SARS-CoV-2-positive and -negative samples, the coefficient of variation (CV) for measuring imprecision was less than 20%. Indeed, CV values above 20% typically indicate adequate reproducibility for enzymatic immunoassays. However, if there is evidence of excessive variation (>30%) within assay runs, further preliminary studies should be performed to verify whether a stabilization of the assay is possible or whether the assay format should be abandoned [45]. RI values did not vary between measurements, indicating high stability during the reactions. Furthermore, no adverse effects were detected when evaluating samples for both SARS-CoV-2-positive and -negative cases across a range of bilirubin, hemoglobin, and triglyceride concentrations. This indicates that the kit is suitable for use in the presence of other medical conditions that may overlap with COVID-19.

Timely and accurate diagnostic testing for SARS-CoV-2 is an essential component of a comprehensive COVID-19 response strategy. Therefore, we investigated the behavior of the IBMP ELISA IgA/IgM/IgG COVID-19 and other commercial tests during seroconversion of COVID-19. The IBMP ELISA IgA/IgM/IgG COVID-19, GOLD ELISA COVID-19 IgG/IgM, and Anti-SARS-CoV-2 IgG ELISA detect seroconversion in the first week post symptom onset. In fact, the humoral response to SARS-CoV-2 can aid in the diagnosis of COVID-19, including asymptomatic or subclinical cases [14], where infected individuals should be treated with non-pharmacological measures to prevent the spread of SARS-CoV-2. Surprisingly, the Anti-SARS-CoV-2 IgM ELISA detects seroconversion on day 9 post the onset of symptoms, which is likely due to antigen preparation. Given the rapid recovery observed in most patients infected with recent SARS-CoV-2 variants by day 6, the detection timeline of the IBMP ELISA IgA/IgM/IgG COVID-19, which identifies seroconversion at 6 days post onset (in contrast to the GOLD ELISA COVID-19 IgG/IgM at 4–5 days), may not be sufficiently prompt to inform treatment strategies in acute cases. This is particularly critical for patients with potentially severe disease, such as those with immunodeficiencies or cancer, where earlier diagnosis via PCR tests is essential. While antibody tests provide valuable insights for retrospective analysis, population-level surveillance, and assessing immune responses post infection or post vaccination, their use should be complemented with molecular diagnostics to ensure timely and effective patient management.

## 5. Strengths and Limitations

The main limitation of the present study is the small number of patients studied. However, the study was performed according to the STARD protocol, and the number of samples was calculated statistically. Further studies with a larger number of samples will serve to elucidate and confirm the results described here. Another limitation is the evaluation of commercial assays using only SARS-CoV-2-positive samples, which limits diagnostic performance to the determination of specificity. However, our goal was to evaluate the diagnostic performance of the IBMP ELISA IgA/IgM/IgG COVID-19 kit, and the commercial assays were only used to compare the ability of the IBMP kit to identify the disease in individuals who are positive for COVID-19.

## 6. Conclusions

The results obtained so far with the IBMP ELISA IgA/IgM/IgG COVID-19 kit show that it is able to distinguish positive from negative samples for COVID-19 and has a high diagnostic capacity as the disease progresses. The IBMP ELISA IgA/IgM/IgG COVID-19 kit has similar or better diagnostic performance than the commercial assays investigated in this study, high intra-assay reproducibility, and a high detection of seroconversion at an early stage of the infection.

## Figures and Tables

**Figure 1 diagnostics-14-01514-f001:**
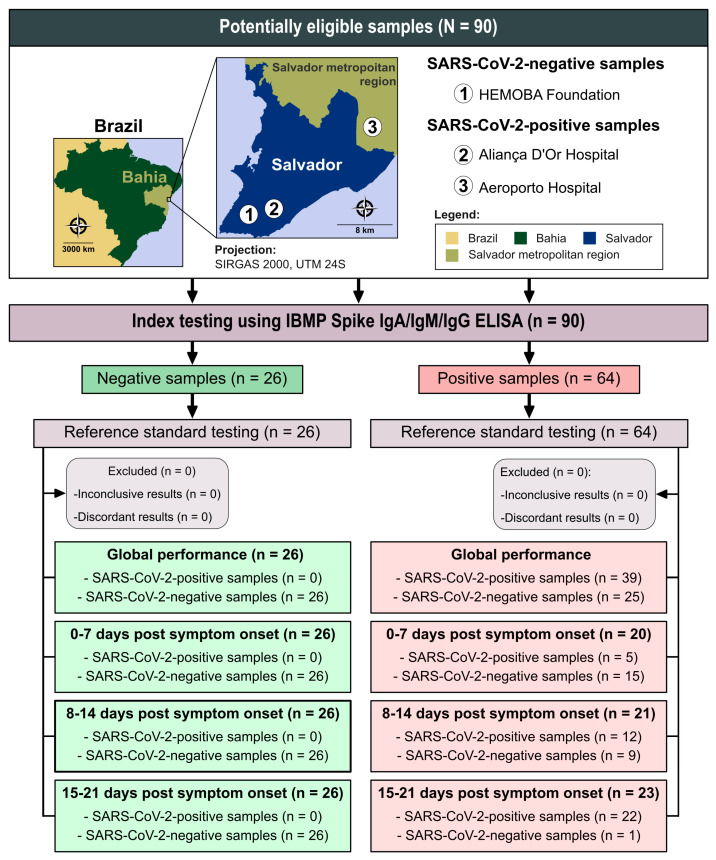
Flowchart illustrating study design in conformity with the Standards for Reporting of Diagnostic Accuracy Studies (STARD) guidelines. Source base layer and credit base layer: https://data.humdata.org/ (accessed on 14 April 2024) under creative commons attribution for intergovernmental organizations: https://data.humdata.org/dataset/geoboundaries-admin-boundaries-for-brazil (accessed on 14 April 2024).

**Figure 2 diagnostics-14-01514-f002:**
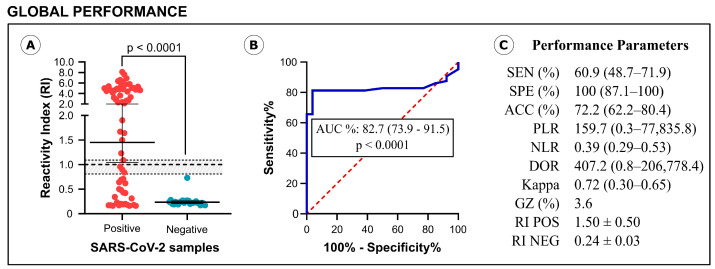
Graphical analysis of reactivity index (**A**) and area under the ROC curve (AUC) (**B**) obtained with serum samples from SARS-CoV-2-positive and SARS-CoV-2-negative samples. The cut-off value is the reactivity index = 1.0 and the shaded area represents the gray zone (0.80 ≤ IR < 1.10). Performance parameters (**C**) obtained for IBMP Spike IgA/IgM/IgG ELISA. SEN, sensitivity; SPE, specificity; ACC, accuracy; PLR, positive likelihood ratio; NLR, negative likelihood ratio; DOR, diagnostic odds ratio; Kappa, Cohen’s Kappa coefficient; GR, gray zone; RI, reactivity index; POS, positive; NEG, negative.

**Figure 3 diagnostics-14-01514-f003:**
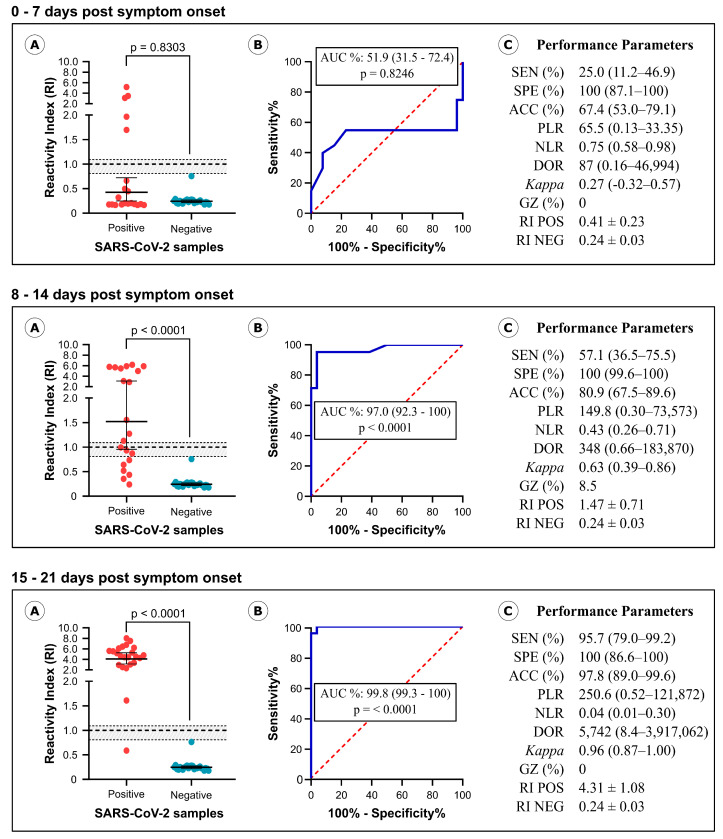
Analysis of SARS-CoV-2-positive samples classified by symptom onset. Graphical analysis of reactivity index (**A**) and area under the ROC curve (AUC) (**B**) determined using serum samples from SARS-CoV-2 positive and SARS-CoV-2-negative samples. The cut-off value is the reactivity index = 1.0 and the shaded area represents the gray zone (0.80 ≤ IR < 1.10). Performance parameters determined for the IBMP Spike IgA/IgM/IgG ELISA (**C**). SEN, sensitivity; SPE, specificity; ACC, accuracy; PLR, positive likelihood ratio; NLR, negative likelihood ratio; DOR, diagnostic odds ratio; Kappa, Cohen’s Kappa coefficient; GZ, gray zone; RI, reactivity index; POS, positive; NEG, negative.

**Figure 4 diagnostics-14-01514-f004:**
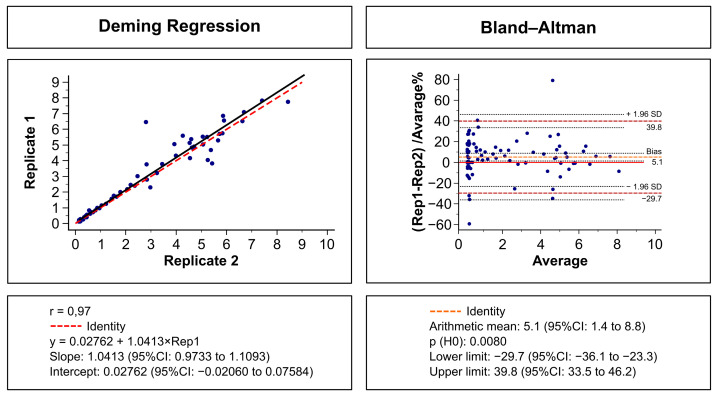
Deming regression fit (left) and Bland–Altman plots (right) comparing two replicates (Rep 1 and Rep 2) of the IBMP Spike IgA/IgM/IgG ELISA for the detection of anti-SARS-CoV-2 using SARS-CoV-2-positive (*n* = 64) and -negative (*n* = 26) samples.

**Figure 5 diagnostics-14-01514-f005:**
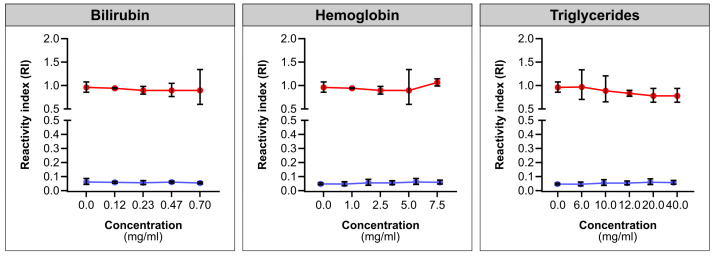
Interference of bilirubin, hemoglobin, and triglycerides on the IBMP Spike IgA/IgM/IgG ELISA signal. Red line (positive samples); Blue line (negatieg samples).

**Figure 6 diagnostics-14-01514-f006:**
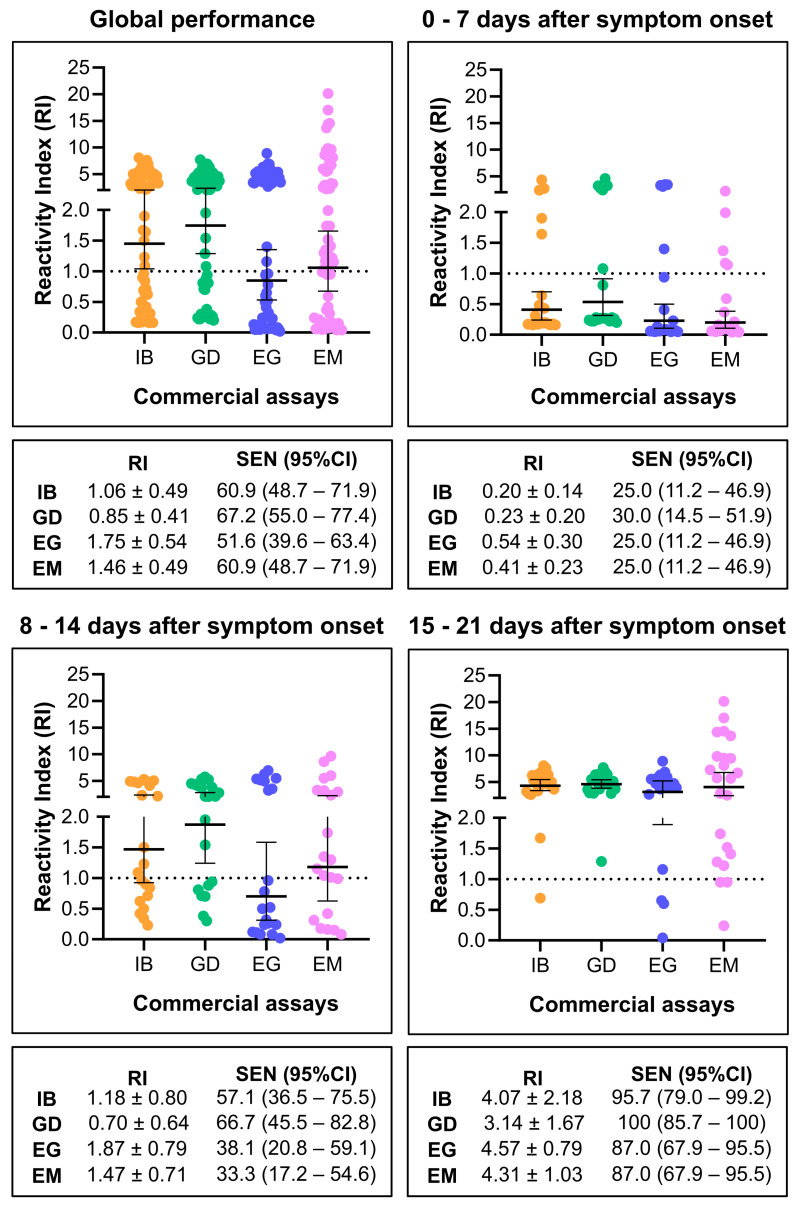
Comparison of the reactivity index (RI) and sensitivity (SEN) of the IBMP Spike IgA/IgM/IgG ELISA with three available commercial kits. IB, IBMP Spike IgA/IgM/IgG ELISA; GD, GOLD ELISA COVID-19 IgG/IgM; EG, Anti-SARS-CoV-2 IgG ELISA; EM, Anti-SARS-CoV-2 NCP IgM ELISA.

**Figure 7 diagnostics-14-01514-f007:**
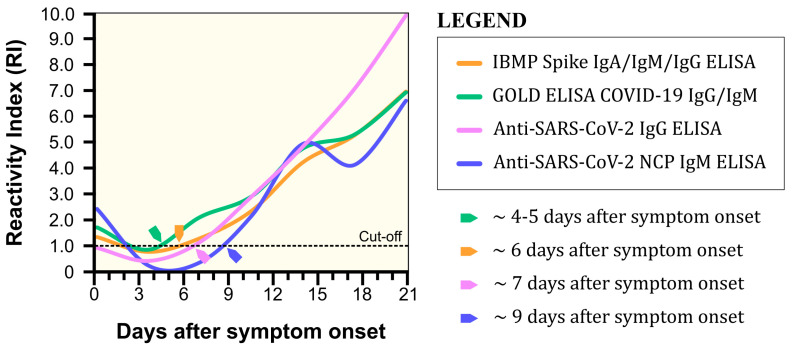
Seroconversion of COVID-19 by IBMP Spike IgA/IgM/IgG ELISA and three commercial assays in serum samples obtained at different time points post symptom onset.

**Table 1 diagnostics-14-01514-t001:** Intra-lab testing: repeatability, between-runs, within-days, and within-laboratory evaluations.

	Samples	Repeatability		Between Runs		Within Days		Within Lab	
		SD	CV%	SD	CV%	SD	CV%	SD	CV%
Batch 1	#1 (RI = 0.16)	<0.01	2.90	0.06	29.68	0.70	30.48	0.06	25.62
	#2 (RI = 0.16)	<0.01	4.19	0.08	36.32	0.07	27.67	0.06	28.94
	#3 (RI = 1.30)	0.03	10.69	0.12	8.57	0.16	11.48	0.13	9.52
	#4 (RI = 1.50)	0.02	5.74	0.11	7.35	0.16	10.03	0.25	14.69
	#5 (RI = 2.17)	0.02	3.02	0.13	5.81	0.10	4.23	0.25	10.59
	#6 (RI = 4.46)	0.07	6.60	0.02	0.50	0.18	3.88	0.57	12.17
Batch 2	#1 (RI = 0.16)	<0.01	0.26	0.08	47.11	0.07	35.22	0.10	40.98
	#2 (RI = 0.16)	<0.01	2.51	0.07	41.51	0.07	37.35	0.10	41.73
	#3 (RI = 1.30)	0.01	2.41	0.05	3.99	0.05	3.89	0.06	5.40
	#4 (RI = 1.50)	0.02	5.01	0.21	13.69	0.27	16.04	0.29	15.55
	#5 (RI = 2.17)	0.03	4.97	0.42	19.59	0.42	18.29	0.44	17.61
	#6 (RI = 4.46)	0.08	7.86	0.89	22.44	1.03	23.07	0.83	17.82
Batch 3	#1 (RI = 0.16)	<0.01	1.11	0.03	19.24	0.08	36.98	0.06	27.38
	#2 (RI = 0.16)	<0.01	1.94	0.02	12.23	0.06	28.24	0.06	25.61
	#3 (RI = 1.30)	0.01	5.37	0.17	12.93	0.15	11.47	0.12	9.49
	#4 (RI = 1.50)	0.02	6.55	0.42	23.93	0.36	19.14	0.34	18.17
	#5 (RI = 2.17)	<0.01	0.11	0.39	16.86	0.36	14.69	0.37	15.92
	#6 (RI = 4.46)	0.01	0.62	0.85	18.70	0.92	18.75	0.75	15.41

RI (reactivity index) calculated using the Anti-SARS-CoV-2 NCP IgG ELISA kit from Euroimmun Medizinische Labordiagnostika AG, Lübeck, Germany. SD (Standard deviation); CV% (Coefficient of variation). SARS-CoV-negative samples (#1 and #2); SARS-CoV-2-positive samples (#3 to #6).

## Data Availability

The raw data supporting the conclusions of this article are provided by the authors without reservation.

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
