# Peer review of "Comprehensive Study of the IBMP ELISA IgA/IgM/IgG COVID-19 Kit for SARS-CoV-2 Antibody Detection"

_diagnostics, 2024, doi:10.3390/diagnostics14141514_

Round 1

Reviewer 1 Report

Comments and Suggestions for Authors

Thank you for an opportunity to review this article. The article provides relevant information on the diagnosis of COVID-19 and is relevant to healthcare workers, policy makers, as well as the general public. The introduction orients the reader to the topic satisfactorily. The authors also clearly stated the aim of the study in the introduction. The materials and methods section clearly explains how the study was conducted, making it possible for other researchers to replicate the study. The findings are clearly presented, with the help of clear tables and figures. The discussion is robust and the authors also discuss the strengths and limitations of their study. The conclusion is based on the study findings. I have a few suggestions to further strengthen the paper.

MINOR REVISIONS

1. Keywords should be in alphabetic order.

2. Under materials and methods, the first subsection should be ‘study design’ where authors explain the study design used.

3. The results section can be subdivided into 2 sections, for example, ‘characteristics of participants’, and ‘diagnostic performance OF IBMP Spike IgA/IgM/IgG ELISA’

4. The limitations section should be titled ‘Strengths and limitations’, and it should come before the conclusion.

5. In lines 423-426, the authors repeated results as they are in the results section. They should just summarise the findings.

6. In lines 444-452, the authors repeated results as they are in the results section. They should just summarise the findings.

7. In line 513, the authors state, ‘….samples was calculated statically.’ Change ‘statically’ to ‘statistically’

Comments on the Quality of English Language

Minor editing required.

Author Response

Comment #1. Keywords should be in alphabetic order.

 Reply: We extend our gratitude to the reviewer for highlighting this issue. We have revised the keywords to be in alphabetical order:

Before: “SARS-CoV-2; COVID-19; Immunodiagnosis; Sensitivity; Specificity.”

After (line 48): “COVID-19; Immunodiagnosis; SARS-CoV-2; Sensitivity; Specificity.”

Comment #2. Under materials and methods, the first subsection should be ‘study design’ where authors explain the study design used.

 Reply: Thank you for bringing this to our attention. We have revised the text as follows:

Before: “2.1. Sample size and collection of human serum”.

After (line 96): “2.1. Study design”.

Comment #3. The results section can be subdivided into 2 sections, for example, ‘characteristics of participants’, and ‘diagnostic performance OF IBMP Spike IgA/IgM/IgG ELISA’.

 Reply: Thank you for bringing this to our attention. We have added the new two subsection as suggested:

Before: No subsections.

After       (line 223): “3.1. Characteristics of participants”

                (line 237): “3.2. Diagnostic performance of IBMP Spike IgA/IgM/IgG ELISA”

Comment #4. The limitations section should be titled ‘Strengths and limitations’, and it should come before the conclusion.

Reply: As per the reviewer's request, we have replaced "Limitations" with "Strengths and limitations" and placed this subsection before the “Conclusions”:

Before: “6. Limitations”

After (line 507): “5. Strengths and limitations”

Comment #5. In lines 423-426, the authors repeated results as they are in the results section. They should just summarise the findings.

 Reply: We thank the reviewer for pointing out the repetition of results in the discussion section. However, the inclusion of this repetitive text was intentional to facilitate reading and to prevent the reader from having to refer back to the Results section to recall the data. Given the extensive amount of data we explored, we believe it is crucial that the text remains unchanged to ensure a smooth reading experience. Therefore, we kindly ask the reviewer to reconsider this critique.

Comment #6. In lines 444-452, the authors repeated results as they are in the results section. They should just summarise the findings.

 Reply: We thank the reviewer for pointing out the repetition of results in the discussion section. However, the inclusion of this repetitive text was intentional to facilitate reading and to prevent the reader from having to refer back to the Results section to recall the data. Given the extensive amount of data we explored, we believe it is crucial that the text remains unchanged to ensure a smooth reading experience. Therefore, we kindly ask the reviewer to reconsider this critique.

Comment #7. In line 513, the authors state, ‘…samples was calculated statically.’ Change ‘statically’ to ‘statistically’.

 Reply: Thank you for bringing this to our attention. We have corrected the text:

Before: “…samples was calculated statically. Further studies…”

After (line 510): “…samples was calculated statistically. Further studies…”

Reviewer 2 Report

Comments and Suggestions for Authors

Dear authors,

I have reviewed the manuscript "Comprehensive Study of the IBMP ELISA IgA/IgM/IgG..." and find it to be well-written and addressing a relevant topic. The methodologies described are clear and align well with the reported results. The presentation of results is adequate and supports the authors' discussions. The references cited are up-to-date.

I recommend that the authors provide a discussion on the rationale for including IgA detection in the kit. What motivated this choice, and how does it contribute to the detection system? Additionally, it would be beneficial for the authors to discuss the selection of the S1 domain as the focus of the study. What are the advantages and limitations of this choice?

 Best regards,

Author Response

Comment #1. I recommend that the authors provide a discussion on the rationale for including IgA detection in the kit. What motivated this choice, and how does it contribute to the detection system? Additionally, it would be beneficial for the authors to discuss the selection of the S1 domain as the focus of the study. What are the advantages and limitations of this choice?

 Reply: Thank you for bringing this to our attention. The IBMP Spike IgA/IgM/IgG ELISA employs the whole spike molecule, not only the S1 subunit as mentioned by the reviewer. This information is already available in lines 412-414, as described below:

“In contrast, the IBMP ELISA IgA/IgM/IgG COVID-19 kit uses only one recombinant protein (spike, whole molecule) to detect IgG and IgM antibodies in addition to IgA (not used in the other kits evaluated here).”

Conversely, two other commercial kits use the S1 subunit as an antigen (lines 170-175):

“…(2) Anti-SARS-CoV-2 IgG ELISA (Euroimmun Medizinische Labordiagnostika AG, Lübeck, Germany), which detects IgG antibodies to SARS-CoV-2 using the S1 domain of the spike protein, including the immunologically relevant receptor binding domain (RBD);  and (3) GOLD ELISA COVID-19 IgG + IgM (REM Diagnóstica, São Paulo-SP, Brazil), which detects both IgG and IgM using the S1 and S2 domains of the spike protein and the N protein.”

The developers introduced the possibility of detecting anti-spike IgA (whole molecule) to differentiate the kit from others commercially available, as the IBMP ELISA IgA/IgM/IgG COVID-19 kit can also detect specific antibodies in breast milk. However, we did not evaluate this application in our study. To clarify this potential use in our manuscript, we have added the following sentence to the discussion section:

(Line 414-415): “The presence of IgA in human milk suggests that this biological material could serve as an alternative for COVID-19 diagnosis [37,38].”

Hence, we have included two new references:

(Line 656-658): Fox, A.; Marino, J.; Amanat, F.; Krammer, F.; Hahn-Holbrook, J.; Zolla-Pazner, S.; Powell, R.L. Robust and Specific Secretory IgA against SARS-CoV-2 Detected in Human Milk. iScience 2020, 23, 101735, doi:10.1016/j.isci.2020.101735.

(Line 659-660): Rio-Aige, K.; Azagra-Boronat, I.; Castell, M.; Selma-Royo, M.; Collado, M.C.; Rodríguez-Lagunas, M.J.; Pérez-Cano, F.J. The Breast Milk Immunoglobulinome. Nutrients 2021, 13, 1810, doi:10.3390/nu13061810.

Reviewer 3 Report

Comments and Suggestions for Authors

Thank you for sharing your article on a comprehensive study of the IBMP ELISA IgA/AgM/IgG COVID-19 kit for the detection of SARS-CoV-2 antibodies. Your article reads very well. The just few comments that may help to improve the article:

L101: Is the number of SARS-CoV-2 negative serum samples stated a convenience sample or is it derived from a sample size calculation? Please state in your manuscript.

L110-111: How was the healthy status confirmed and were those subjects also tested for SARS-CoV-2 besides Chagas, HBV, HCV, HIV-1/2, HTLV-1/2 and syphilis? Please revise.

L113: Based on the information provided so far it is not clear what you mean by convenience sample size at this status of the manuscript. Please clarify in your manuscript. 

L128: Which controls were used? Please provide more information. 

Author Response

Comment #1. L101: Is the number of SARS-CoV-2 negative serum samples stated a convenience sample or is it derived from a sample size calculation? Please state in your manuscript.

 Reply: We are grateful for the reviewer's thoughtful input. We would like to address the concern regarding the use of convenience sample size. To evaluate the diagnostic performance of the kit, we used SARS-CoV-2-positive and negative samples determined statistically, as described in the section between lines 97 and 102. For other analyses, such as imprecision, interferents, and cross-reactivity, we employed a convenience sample size, as stated in lines 112-133. In our opinion, the information provided is clear. Therefore, we kindly ask the reviewer to reconsider their critique.

Comment #2. L110-111: How was the healthy status confirmed and were those subjects also tested for SARS-CoV-2 besides Chagas, HBV, HCV, HIV-1/2, HTLV-1/2 and syphilis? Please revise.

 Reply: We appreciate the reviewer's concern regarding the selection of negative samples used in our study. We would like to clarify that the negative samples for SARS-CoV-2 were collected years before the onset of the pandemic. To ensure their inclusion in the study, all samples were evaluated using commercial tests for SARS-CoV-2 and confirmed to be negative. Additionally, this sample set was tested and found negative for Chagas disease, HBV, HCV, HIV-1/2, HTLV-1/2, and syphilis, thereby certifying their status as eligible blood donors.

Comment #3. L113: Based on the information provided so far it is not clear what you mean by convenience sample size at this status of the manuscript. Please clarify in your manuscript.

 Reply: We are grateful for the reviewer's thoughtful input. We would like to address the concern regarding the use of convenience sample size. To evaluate the diagnostic performance of the kit, we used SARS-CoV-2-positive and negative samples determined statistically, as described in the section between lines 97 and 102. For other analyses, such as imprecision, interferents, and cross-reactivity, we employed a convenience sample size, as stated in lines 112-133. In our opinion, the information provided is clear. Therefore, we kindly ask the reviewer to reconsider their critique.

Comment #3. L128: Which controls were used? Please provide more information.

 Reply: We appreciate the reviewer's valuable insight. The IBMP Spike IgA/IgM/IgG ELISA is a commercial kit produced by the Molecular Biology Institute of Paraná, a Brazilian technology company. The kit includes calibrators and controls provided by the manufacturer, along with other components such as sensitized microplates and reagents. All steps were conducted according to the manufacturer's instructions.
